# Assessment of Factors Affecting Pavement Rutting in Pakistan Using Finite Element Method and Machine Learning Models

**Xiao Hu, Azher Ishaq *, Afaq Khattak * and Feng Chen**

The Key Laboratory of Road and Traffic Engineering, Ministry of Education, Tongji University, Shanghai 201804, China; hux@suog.cn (X.H.); fengchen@tongji.edu.cn (F.C.)
* Correspondence: azherishaq@tongji.edu.cn (A.I.); khattak@tongji.edu.cn (A.K.)

**Abstract:** This study researches environmental factors, vehicle dynamics, and loading conditions on pavement structures, aiming to comprehend and predict their impact. The susceptibility of asphalt pavement to temperature variations, vehicle speed, and loading cycles is explored, with a particular focus on the lateral distribution of wheel tracks in driving and passing lanes. Utilizing video analysis and finite element modelling (FEM) through ABAQUS 2022 software, multiple input factors, such as speed (60, 80 and 100 km/h), loading cycles (100,000 to 500,000), and temperature range (0 °C to 50 °C), are applied to observe the maximum rutting (17.89 mm to 24.7 mm). It is observed that the rut depth exhibited is directly proportional to the loading cycles and temperature, but the opposite is true in the case of vehicle speed. Moreover, interpretable machine learning models, particularly the Bayesian-optimized light gradient boosting machine (LGBM) model, demonstrate superior predictive performance in rut depth. Insights from SHAP interpretation highlight the significant roles of temperature and loading frequency in pavement deformation. This study concludes with a comprehensive understanding of how these factors impact road structures in Pakistan. Its implications extend to valuable insights for optimizing road design, offering a significant contribution to enhancing the durability and sustainability of road infrastructure in the region.

**Keywords:** temperature effects; lateral distribution of wheel loads; rutting; machine learning; finite element modelling (FEM)

## 1. Introduction

Road structure is an important component of a well-planned transportation system. It affects how well roads are used for transportation purposes, how well vehicles move along them, and how safe they are for drivers to use [1]. The structure of roads is impacted by various factors, from environmental conditions to the availability of materials, each having a different influence on road structure [1,2]. Among these, the climatic conditions of an area, including environmental temperature; humidity; groundwater levels; and wind speed, can have a significant impact on the type of road structure [3,4]. Traditional ways of transportation and old vehicular network systems are not enough to accommodate the new system for traffic—a privacy preserving reputation updating (PPRU) scheme for cloud-assisted vehicular networks introduced to manage traffic safety [5,6]. In the world of pavement management, early detection of rutting is essential for both preventative maintenance and the development of rehabilitation plans [7]. Ruts—longitudinal depressions in the road surface—emerge along the wheel tracks of vehicles and are typically followed by upheavals along the rut's sides. Hydroplaning and structural collapse are also possible results. Over the course of a pavement's lifetime, rutting develops when permanent deformations collect because of repetitive load applications. Rutting affects the whole pavement structure and causes permanent deformations in all its layers [8]. Analysis of the pavement surface profile helps pinpoint the layer at fault for rutting failure [9].

### 1.1. Effects of Temperature

The effects of temperature are significant in governing the strength of bituminous paving mix. High temperatures pose a significant threat to the durability of asphalt pavement, as it increases the likelihood of rutting or permanent deformation [10]. If thermal conditions are not addressed, they can lead to serious issues. Changes in temperature not only result in the expansion or contraction of materials, but they also have a significant impact on the viscoelastic properties of asphalt mixtures used in pavements [11]. The greater the temperature fluctuation, the higher the level of stress on the pavement [12]. Desouza [13] ran a simulation to study the progression of cracks in asphaltic mixes when they were loaded in a monotonic fashion. In the research carried out by Lutif [14], a multi-scale modelling was used to provide predictions about the mechanical behaviour of asphalt mixes. The temperature fluctuation of asphalt pavements has been the subject of several investigations [15], the findings of which have been published in several academic journals. When designing asphalt roadways, it is essential to have a solid understanding of the link that exists between varying temperatures and the mechanical performance of the asphalt.

### 1.2. Effects of Wheel Load

When subjected to the forces exerted by vehicular traffic, asphalt pavement exhibits dynamic and flexible characteristics. When moving vehicles generate load on pavement structures it is called dynamic behavior [16]. Numerous factors contribute to asphalt pavement, including the quality of the road layers, the features of the traffic location, and the usual conditions of the environment. Investigating the load-bearing capacity of asphalt pavement under mobile loads, Liu [17] employed a semianalytical finite element technique. Cui [18] utilized Biot's theory and Kirchhoff's theory of thin plate to define the dynamic behavior of unsaturated poroelastic ground beneath uneven pavement. Gao [19] introduced novel fatigue reliability models for composite materials, considering the combined effects of performance degradation, effective stress growth, and uncertainties in initial strength and stiffness, thereby offering a comprehensive approach to assess fatigue damage and failure thresholds. Gao [20] also worked to establish a relationship between fatigue-damaged material and fatigue life using the adaptive neuro-fuzzy inference system (ANFIS).

### 1.3. Effects of Vehicle Speed

The acceleration field of pavement when under a moving load was analyzed numerically by Yan [21]. Sensitivity analysis revealed that the acceleration response of intact pavement is primarily influenced by vehicle speed, with vehicle load following as the next significant factor. The impact attributed to the elastic modulus of the upper layer was comparatively modest. To encapsulate this response, Yan's proposal introduced the effective amplitude of vertical acceleration. Liu [17] utilized a discrete-continuous coupling simulation to examine the dynamic stress responses of asphalt pavement under moving wheel loads. The findings revealed that as the magnitudes and velocities of the loads heightened, the impact load generated by the wheel loads increased, accompanied by a corresponding rise in both stress wave energy and wave peak. According to Khavandi [22], the pavement surface's greatest rut depth was in the pavement center (passing lane), due to the effect of the rear tandem axles of vehicles over the center portion with the passage of time. Also, vehicle speed affects rut depth more than weight. Three-axle vehicles have a bigger influence than two-axle vehicles.

### 1.4. Summary and Structure of the Paper

Section 1 describes the recent research on fatigue damage and environmental effects on road structures. Different types of vehicles load and vehicle speed affect roads. An overview of the content of this study is presented and the broader research landscapes of related fields are introduced. The rest of the paper is organized as follows: in Section 2, the materials and methodology of the paper is described. Research design, data collection procedures, and analytical methods are core points in this section. Section 3 holds the

results and findings of the study, and Section 4 contains a discussion of the results and their comparison with previous research, and the conclusion.

## 2. Materials and Methodology

### 2.1. Study Area

The road cross-section shown in Figure 1 represents the upgradation of an existing road between two cities, Lahore to Gujranwala, taken from the Communication and Works Department of Pakistan. The total length of this road is around 100 km. The thickness of layers such as the asphalt wearing course, asphalt concrete base course, water bound macadam (WBM) and granular subbase course is 5 cm, 8 cm, 30 cm, and 20 cm, respectively.

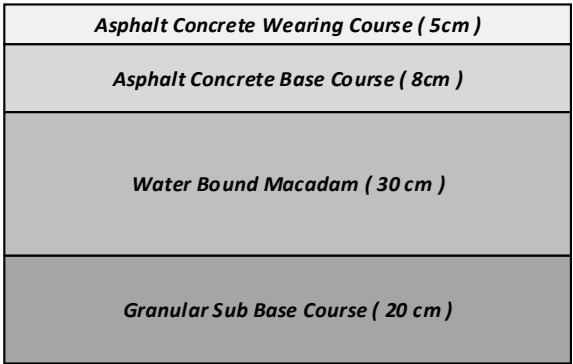

**Figure 1.** Asphalt pavement design.

### 2.2. FEM Analysis

FEM software ABAQUS 2022, used in this investigation, is an effective tool that makes it simple to find solutions to difficult issues by using nonlinear analysis. The program has a library that contains a wide variety of material behavior models and a significant number of discrete finite elements. ABAQUS is an effective modelling tool that may be used to determine whether there is a link between the specification of a hot mix asphalt's transverse section and the rutting distribution of the asphalt [23]. In recent years, there has been a rise in the use of FEM to analyze the structure of pavements. This is because a nonlinear link between stress and various forms of stresses can be readily established [24]. Figure 2 shows the workflow in ABAQUS. The diagram outlines the procedural flow, starting with the definition of the model geometry, material properties, and boundary conditions. Subsequently, users specify the analysis type, such as static or dynamic, mesh modelling, and configure solver settings. The simulation is then executed, and results are postprocessed to extract relevant data.

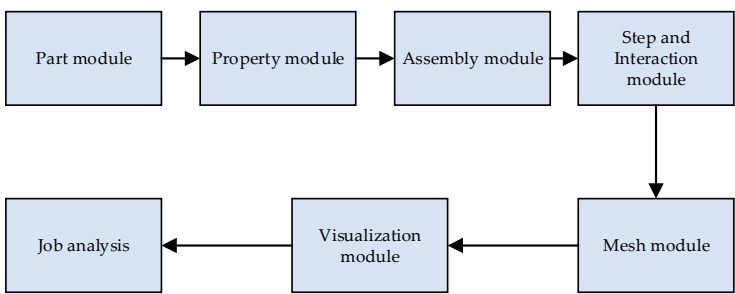

**Figure 2.** Workflow diagram in ABAQUS.

### 2.3. Input Parameters and Methodology

In this research, property module was used to define materials and to define sections. The heat transfer mechanical model requires specific properties, which are referred to as temperature-dependent speed parameters. This means that the density, conductivity, spe-

cific heat, Young's Modulus, Poisson ratio, and creep properties of the materials can fluctuate at varying temperatures. The specific values are shown in Table 1.

**Table 1.** Material properties.

| Parameter | Asphalt | Cement Stabilized Base | Lime Soil Subbase | Soil Foundation |
|---|---|---|---|---|
| Thermal conductivity ($\lambda$) | 4680 | 5616 | 5148 | 5616 |
| Density ($\rho$) | 2300 | 2200 | 2100 | 1800 |
| Heat capacity (J/K) | 924.9 | 911.7 | 942.9 | 1040.0 |
| Solar absorption rate | 0.9 | | | |
| Road surface emissivity | 0.81 | | | |
| Absolute zero (°C) | $-273$ | | | |
| Stefan–Boltz constant ($\sigma$) | $2.041092 \times 10^{-4}$ | | | |

To reduce the time required for FE analysis computations, the effect of repeated loading on asphalt pavement rutting was reduced to a single loading step. Equation (1) was used for the calculation of wheel load cumulative action times [25] as follows:

$$t = 0.36NP/(n_w pBv) \tag{1}$$

where $t$ is the cumulative action time of the wheel load; $N$ is number of action times of the wheel load; $P$ is the axle weight of the vehicle; $n_w$ denotes the number of wheels; $p$ denotes the ground pressure of the tire; $B$ is the tire ground width; and $v$ refers to speed.

All parameters are considered constant values (shown in Table 2) to investigate the impact of temperature. According to Equation (1), the duration of a single action time for axle loads is 0.008641 s. This equation was chosen as it has been widely accepted for determining one-time action time. To validate the proposed solution, three distinct scenarios were tested. Each scenario had a unique action time and load times, and was tested on different temperatures and vehicle speed.

**Table 2.** Input Parameters.

| No. | Input Parameter | Value |
|---|---|---|
| 1 | Number of wheel loads, times | 1 |
| 2 | Vehicle axle load, kN | 100 |
| 3 | Wheel number of the shaft, pcs | 4 |
| 4 | Tire ground pressure, MPa | 0.7 |
| 5 | Tire contact width, cm | 18.6 |
| 6 | Travel speed, km/h | 100, 80, 60 |

The simulation was performed selecting different scenarios depending on the vehicle speed and action time. Each scenario was simulated multiple times to check the effect of different input parameters, such as tire ground pressure, number of wheels, temperatures, and vehicle speed. Equation (1) was used to calculate the duration of a single time for axle loads. Table 3 shows the model simulation scenario adopted for this research.

*2.4. Machine Learning Techniques*

In this research, the prediction of rutting intensity was calculated utilizing four distinct machine learning regression algorithms: adaptive boosting (AdaBoost), random forest (RF), linear regression (LR), and light gradient boosting machine (LGBM). The data employed for this analysis was sourced from the Communication and Works Department of Pakistan. The optimization of hyperparameters in the machine learning regression models

was executed through Bayesian optimization. Figure 3 illustrates the development process of the machine learning regression model.

**Table 3.** Model Scenario Development.

| Scenario | Description |
|---|---|
| Scenario 1 | With speed limit 100 km/h and 100,000 to 500,000 action times with 0 °C to 50 °C temperature range |
| Scenario 2 | With speed limit 80 km/h and 100,000 to 500,000 action times with 0 °C to 50 °C temperature range |
| Scenario 3 | With Speed limit 60 km/h and 100,000 to 500,000 action times with 0 °C to 50 °C temperature range |

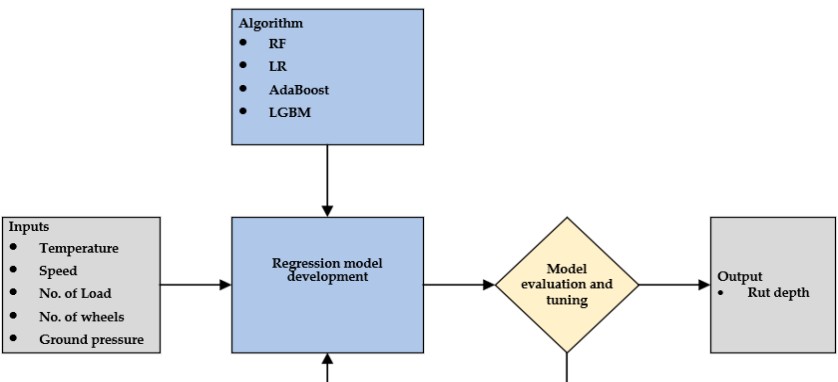

**Figure 3.** Diagram illustrating machine learning models, depicting input and output variables.

To anticipate rutting intensity based on the input factors listed in Table 4, it is important to highlight that label encoding was implemented before constructing the predictive models. Large vehicle such as trucks with different numbers of wheel (4, 8, and 12) were considered in this experiment.

**Table 4.** Range of factors.

| Factors | Data Type | Range |
|---|---|---|
| Temperature (T) | Discrete | 0, 10, 20, 30, 40, 50 |
| Speed (v) | Discrete | 60, 80, 100 |
| No. of Load (N) | Discrete | 100,000, 200,000, 300,000, 400,000, 500,000 |
| No. of wheels ($n_w$) | Discrete | 4, 8, 12 |
| Ground pressure (p) | Discrete | 0.7, 0.9, 1.1 |

### 2.4.1. Random Forest (RF) Regression

The RF consists of multiple predictors based on trees, with each tree trained using values derived from an independently sampled and randomly generated path. These paths share a common distribution across all trees within the forest. Specifically, the kth tree undergoes training using a unique random path denoted as $\zeta_k$, which adheres to the same circulation as the previously mentioned arbitrary paths. Accordingly, a tree $\Psi$ (X, $\zeta_k$) is formed, with X representing the input path. By combining the average predictions from numerous trees in the forest, the RF enhances predictive accuracy and addresses concerns related to overfitting. The mathematical representation is expressed in Equation (2) as follows:

$$\hat{y} = \frac{1}{l} \sum_{k=1}^{l} \psi_k(x) \tag{2}$$

In this context, $\hat{y}$ represents the predicted output, and $l$ stands for the total number of generated trees (where $1 \leq k \leq l$). The mean-squared error for any tree, denoted as

$\psi(X)$, is expressed as $E[X, Y](Y - \psi(X))^2$, where *X* is the input and *Y* is the output path. Equation (3) illustrates the mean-squared simplification as the number of trees in the forest approaches infinity.

$$E_{X,Y}(Y - \Lambda_{k\psi}(X, \zeta k))^2 \rightarrow E_{X.Y}(Y - E_{\zeta\psi}(X, \zeta))^2 \qquad (3)$$

It is essential to highlight that the training dataset in RF was created through the bagging technique, incorporating the addition of bootstrap illustrations, as shown in Figure 4.

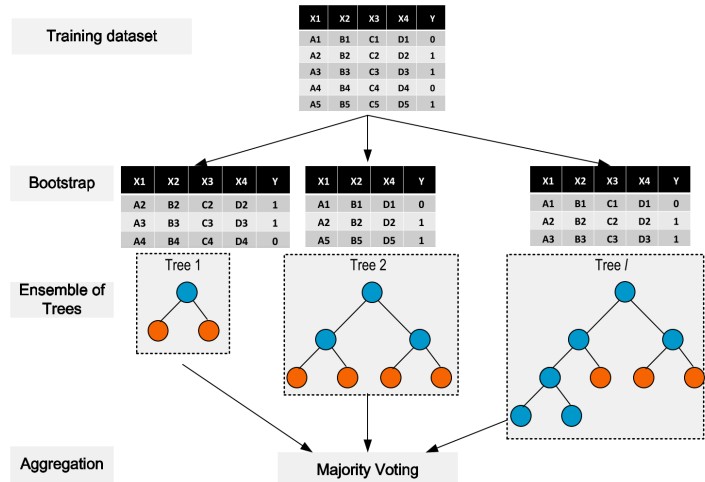

**Figure 4.** RF regression model.

### 2.4.2. Light Gradient Boosting Machine (LGBM) Regression

LGBM is a type of learning framework that combines boosting and decision trees, specifically falling under the category of gradient learning. Differing from the RF model, LGBM uses histogram-based strategies to minimize memory usage, choosing a leafwise expansion (LWE) approach with depth constraints to increase the training phase. This requires constructing a histogram with a width limit, $\gamma$, representing the number of boxes for dividing continuous floating-point variable values. These histograms are stored with 8-bit integers, reducing memory requirements to 1/8 of the original size by removing the necessity for extra storage of presorted results. Remarkably, the model's accuracy remains unaffected despite this imprecise partitioning.

Within the LGBM model, the accuracy of segmentation points in the decision tree holds less significance, as the tree itself acts as a regularization mechanism, mitigating the risk of overfitting. Figure 5 illustrates a graphical representation of the node and leaf levels in LGBM and LGBM tree expansions.

### 2.4.3. Adaptive Boosting (AdaBoost) Regression

The AdaBoost algorithm is a clear and simple method that constructs a robust regressor by combining multiple learning algorithms and finishing in a highly accurate model. Its fundamental concept involves multiple trainings of the dataset to determine the masses of weak regressors, enabling precise predictions for unusual observations and significantly enhancing the ability to forecast future events. The operational code of AdaBoost is outlined as follows:

1. Initially, set the distribution of weight (denoted as $\pi$), where $\pi$ is set to 1/m.
2. Subsequently, using weight distribution at each repetition (t), the weak point is accomplished and represented as ht: $x \rightarrow R$.
3. Consistent with the patterns observed in the training dataset, adjust the distribution of weights as follows:

$$\pi k = (\pi_{(k-1)} \llbracket \exp \rrbracket ^(-\psi kh(xk)))/\Omega$$

4.  The ultimate result across all repetitions t = 1, 2, …, T is returned as follows:

$$f(X) = \sum_{t=1}^{t} \pi_t H_t \, \text{and} \, H(X) = \text{sign}(f(X)).$$

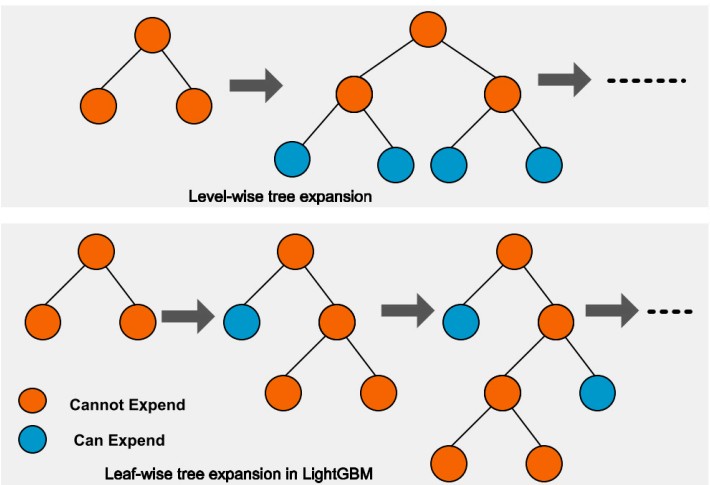

**Figure 5.** Tree expansion in LGBM.

2.4.4. Linear Regression (LR)

Linear regression is a statistical technique used to developed the connection between a dependent variable, (also termed the target or outcome variable), and one or more independent variables (known as predictors or features). The assumption underlying this model is straight, signifying that alterations in the dependent variable are proportionate to variations in the independent variables.

The fundamental structure of a linear regression equation featuring a single independent variable is expressed as follows:

$$y = \beta0 + \beta1 * x + \varepsilon$$

where $y$ signifies the dependent variable (target) to be predicted; $x$ signifies the independent variable (feature) utilized for predictions; $\beta0$ stands for the intercept, indicating $y$ value when $x$ equals 0; $\beta1$ denotes the regression coefficient, portraying the slope of the line; and $\varepsilon$ represents the error in term, showcasing the disparity between real $y$ and forecast $y$.

For scenarios involving multiple independent variables, the linear regression equation extends to

$$y = \beta0 + \beta1 * x1 + \beta2 * x2 + … + \beta n * xn + \varepsilon$$

The primary objective of linear regression is to determine the optimal-fitting line, (or hyperplane in higher dimensions), that reduces the sum of squared errors between projected and definite values. Typically, this process is executed through the least squares method.

2.4.5. Bayesian Optimization for Hyperparameter Tuning

In this study, the modification of hyperparameters within machine learning models was achieved by implementing a Bayesian optimization approach. Widely acknowledged as a global optimization algorithm, this method finds extensive use in various engineering applications. The primary objective involves the optimization of Equation (4), representing the objective function as follows:

$$x\Theta = \frac{argmax \, \varphi(x)}{x \epsilon \Lambda_s} \tag{4}$$

where $x$ represents the hyperparameters; $\Lambda s$ denotes the search space for hyperparameters; and $\varphi(x)$ stands for the objective function, capturing the relationship between the performance of models and their corresponding hyperparameters.

The $R^2$ metric was utilized for evaluation. The objective of this optimization procedure was to identify the most favorable set of hyperparameters ($x\Theta$) that maximizes the model's performance $\varphi(x)$ to its full potential. Bayesian optimization draws the principles of the Bayesian theorem and encompasses the following concepts:

$$\rho(\varphi|T) = \frac{\rho(T|\varphi)\rho(\varphi)}{\rho(T)} \tag{5}$$

where $\varphi$ represent black box function; $\rho(\varphi)$ denotes the prior probability; $\rho(T|\varphi)$ signifies the probability; $\rho(T)$ denotes the normalized constant; and $\rho(\varphi|T)$ represents the posterior probability of $\varphi$.

Utilizing the Gaussian process involves aligning the behavior of the black box function $\varphi$ with the characteristics of a Gaussian distribution. In this scenario, $\Gamma(x)$, which is grounded in the concept of predictable development, assumes the following expression:

$$\Gamma(x) = \begin{cases} (\upsilon(x) - \varphi+)\Lambda(Z) + \alpha(x)\beta(Z) & \alpha(x) > 0 \\ \max(0, \upsilon(x) - \varphi+) & \alpha(x) = 0 \end{cases} \tag{6}$$

$$Z = (\upsilon(x) - \varphi\hat{~}+)/(\alpha(x)) \tag{7}$$

where $\Lambda$ represents the cumulative distribution function, and $\beta$ means the probability density function (PDF).

2.4.6. Performance Measures

The performance metrics of new data obtained using machine learning algorithms can be checked using four different methods.

- Firstly, the mean absolute error (MAE), calculated using Equation (8), which represents the average absolute value of all discrete errors:

$$\text{MAE} = \sum_{x=1}^{\Phi} \frac{|y_x - \hat{y}_x|}{x} \tag{8}$$

- Secondly, the mean squared error (MSE), determined using Equation (9), quantifies model error by computing the difference between observed and predicted values:

$$\text{MSE} = \frac{1}{X}\sum_{x=1}^{\Phi}(y_x - \hat{y}_x)^2 \tag{9}$$

- Thirdly, the root mean squared error (RMSE), presented in Equation (10), is the square root of the mean squared difference:

$$\text{RMSE} = \sqrt{\sum_{x=1}^{\Phi}(y_x - \hat{y}_x)^2} \tag{10}$$

- Finally, the coefficient of determination ($R^2$), as per Equation (11), demonstrates the model's predictive accuracy:

$$R^2 = 1 - \frac{\sum_{x=1}^{\Phi}(y_x - \hat{y}_x)^2}{\sum_{x=1}^{\Phi}\left(y_x - y_{avg}\right)^2} \tag{11}$$

2.4.7. SHAP Interpretation Mechanism

Lundberg and Lee (2018) [26] introduced the SHAP (SHapley Additive exPlanations) mechanism, utilizing game theory principles for post hoc interpretation of machine learning models. SHAP employs an additive factor attribution approach, representing the model

as a linear combination of input factors ($x_1$, $x_2$, ..., $x_r$) for interpretability. Equation (12) is defined as follows:

$$\partial(x') = \Delta_o + \sum_{r=1}^{R} \Delta_r x'_r \tag{12}$$

In the equation, where R denotes input features; $\Delta$ represents Shapley values; and $\Delta o$ is the constant in the absence of input factors, the mapping function $x = \sigma x(x')$ relates inputs $x'$ and $x$. Figure 6 shows the impact of $\Delta 0$, $\Delta 1$, $\Delta 2$, $\Delta 3$ (increasing predicted value), and $\Delta 4$ (decreasing values). Lundberg and Lee discover solutions that are locally accurate, consistent and have no missing information. This ensures that the model's results match the contributions of factors, and changes in influential factors do not reduce their impact. The proposed model is given by Equation (13):

$$\Delta_r(f, \; x) = \sum_{c' \subseteq x'} \frac{|c'|!(\varphi - |c'| - 1)!}{\varphi!} \left[ f_x(c') - f_x(c' \backslash r) \right] \tag{13}$$

where $|c'|$ explains non-zero entries in $c'$ and $c' \subseteq x'$.

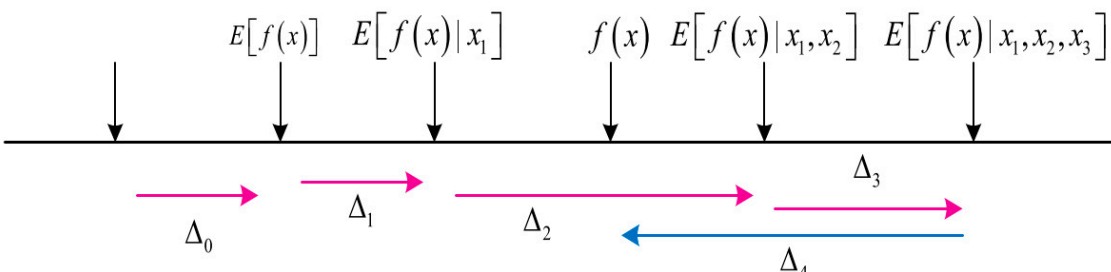

**Figure 6.** SHAP Attributes.

*2.5. Lateral Distribution of Wheel Tracks on Pavement*

2.5.1. Experiment Conditions

The experimental setup involved lanes with a width of 3.75 m for each lane. The total length of the road used in the experiment was 100 km. A closed-circuit television (CCTV) camera was installed on a gantry, with precise adjustments made to its angle and parameters. Video was collected between 7:00 in the morning and 17:00 in the evening, with a total of 168 h and 112 GB data collected in total. During this period, comprehensive video recordings of both the driving and overtaking lanes were gathered. This extensive dataset was then collectively utilized for analysis of the lateral distribution of the vehicle wheel tracks, providing valuable insights into the patterns and behaviour of vehicles on the road. Figure 7 is taken from video.

2.5.2. Method of Data Analysis

To facilitate continuous video recording on specific highway sections, the initial step involved converting captured videos into AVI format files using a computer. These files were subsequently imported into MATLAB for analysis. The analysis focused on selecting the position of the rear wheels on both sides of the vehicle, aiming for the central position. Four predetermined coordinate points guide this selection, enabling coordinate conversion to transform screen coordinates into actual road plane coordinates. This transformation is fundamental for determining wheel track distribution within the lane, obtaining lateral distribution frequencies for diverse vehicle models. Ultimately, the process facilitates the calculation of the standard axle-load track distribution coefficient based on the specific vehicle model under examination.

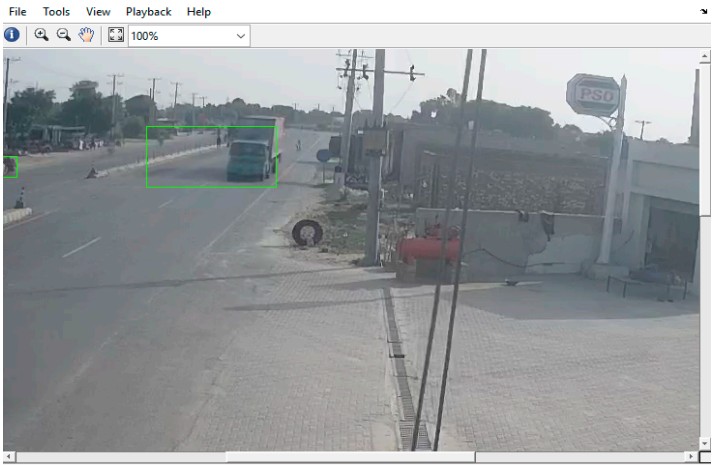

**Figure 7.** Illustration of field test CCTV.

Ghosh [27] proposed a linear transformation method for converting video coordinates to real-world coordinates. The key principle involves calculating a coordinate conversion coefficient, "C", by comparing the screen coordinates and road coordinates of four non-collinear reference points. This coefficient is then used to compute the real-world coordinates of a vehicle's left and right rear wheels from their respective screen coordinates. The accuracy of our analysis greatly depends on this crucial method, emphasizing its significant impact on the overall precision of the process. Gang Wu [28] proposed a detailed solution process for coordinate conversion as follows: Let $x_r$, $y_r$ be road coordinates, $x_s$, $y_s$ be screen coordinates, and $C_1$ to $C_8$ be conversion coefficients. The relationship between the two satisfies

$$x\_r = (C\_1 + C\_2\, x\_s + C\_3\, y\_s)/(C\_4\, x\_s + C\_5\, y\_s + 1) \tag{14}$$

$$y\_r = (C\_6 + C\_7\, x\_s + C\_8\, y\_s)/(C\_4\, x\_s + C\_5\, y\_s + 1) \tag{15}$$

Between them, the conversion coefficient satisfies the four-point non-collinear condition as follows:

$$\begin{bmatrix} C_1 & C_2 & C_3 \\ C_6 & C_7 & C_8 \\ C_4 & C_5 & 1 \end{bmatrix} \neq 0 \tag{16}$$

An equation to solve the conversion coefficient from four known points is as follows:

$$\begin{bmatrix} x_{r,1} \\ x_{r,2} \\ x_{r,3} \\ x_{r,4} \\ y_{r,1} \\ y_{r,2} \\ y_{r,3} \\ y_{r,4} \end{bmatrix} = \begin{bmatrix} 1 & x_{s,1} & y_{s,1} & -x_{s,1}x_{r,1} & -y_{s,1}x_{r,1} & 0 & 0 & 0 \\ 1 & x_{s,2} & y_{s,2} & -x_{s,2}x_{r,2} & -y_{s,2}x_{r,2} & 0 & 0 & 0 \\ 1 & x_{s,3} & y_{s,3} & -x_{s,3}x_{r,3} & -y_{s,3}x_{r,3} & 0 & 0 & 0 \\ 1 & x_{s,4} & y_{s,4} & -x_{s,4}x_{r,4} & -y_{s,4}x_{r,4} & 0 & 0 & 0 \\ 0 & 0 & 0 & -x_{s,1}y_{r,1} & -y_{s,1}y_{r,1} & 1 & x_{s,1} & y_{s,1} \\ 0 & 0 & 0 & -x_{s,2}y_{r,2} & -y_{s,2}y_{r,2} & 1 & x_{s,2} & y_{s,2} \\ 0 & 0 & 0 & -x_{s,3}y_{r,3} & -y_{s,3}y_{r,3} & 1 & x_{s,3} & y_{s,3} \\ 0 & 0 & 0 & -x_{s,4}y_{r,4} & -y_{s,4}y_{r,4} & 1 & x_{s,4} & y_{s,4} \end{bmatrix} \begin{bmatrix} C_1 \\ C_2 \\ C_3 \\ C_4 \\ C_5 \\ C_6 \\ C_7 \\ C_8 \end{bmatrix} \tag{17}$$

After obtaining the conversion coefficient from Equation (17), the road coordinates of any point on the screen can be solved by Equations (14)–(16), and then the position data of both wheels can be obtained. Through the vehicle capture program and coordinate conversion calculation, the position of each vehicle's wheel in the video can be obtained. The process of video analyzing the lateral distribution of vehicle wheel tracks is shown in Figure 8.

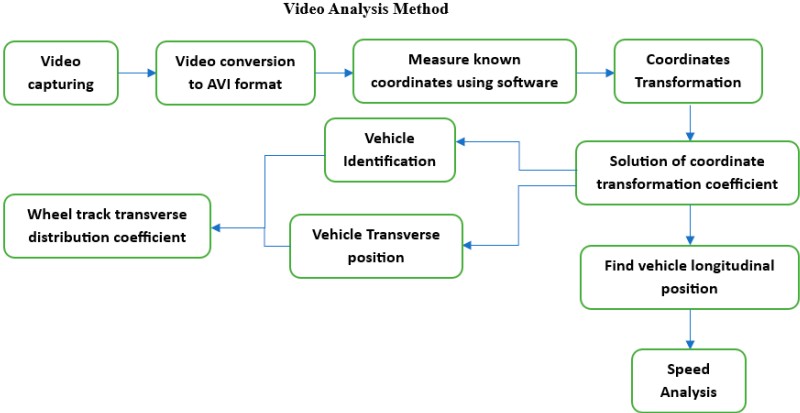

**Figure 8.** Video analysis flowchart methodology.

Statistics on the lateral distribution of wheel tracks generally use the method of dividing the lane into equal-width strips and calculating the frequency of the strips covered by wheel travel. The calculation formula is given by Equation (18),

$$\eta_i = \frac{n_i}{2N} \times 100\% \tag{18}$$

where $\eta_i$ represent the frequency of band $i$(%); $n_i$ denotes the passing ratio on band $i$; and $N$ is total data of passing wheels on the road.

Examining the lateral distribution of wheel tracks entails computing wheel frequency in uniformly spaced bands. The distance is determined from the right-hand side edge line, aligned with traffic flow, with precision ensured through a calibration line strategically positioned in the central area of the intervention-marked test road. This reference line divides the road into 25 transverse bands, each 15 cm wide. These bands, established by the calibration line, serve as essential known coordinate points for subsequent video analysis procedures [28].

$$y = y_o + \frac{A_1}{W_1\sqrt{\frac{\pi}{2}}}\exp\left\{-2\left[\frac{(x - x_{c1})}{W_1}\right]^2\right\} + \frac{A_2}{W_2\sqrt{\frac{\pi}{2}}}\exp\left\{-2\left[\frac{(x - x_{c2})}{W_2}\right]^2\right\} \tag{19}$$

$A_1$ and $A_2$ represent the areas encompassed by the distribution of wheel tracks, while $W_1$ and $W_2$ denote the widths of the concentrated region within the wheel track distribution. The positions of the peak values are designated as $x_{c1}$ and $x_{c2}$.

The representation of wheel mark distribution within a lane is commonly depicted through a frequency diagram, often showcasing a hump-shaped pattern on expressways. However, variations in this distribution arise among different vehicle models, influenced by a diverse traffic composition serving distinct lane functions. To delve deeper, this study focused on driving and overtaking lanes, utilizing observed data from diamond-shaped intervention markings to analyze alterations in wheel track distribution before and after visual intervention for normally traveling vehicles. To ensure analytical accuracy, the dataset excludes instances of lane-changing, focusing solely on collected vehicle data as presented in Table 5.

**Table 5.** Observation data of different type of vehicles from the video.

| Model | Small | Medium | Large Bus | Large Truck | Trailer | Total |
|---|---|---|---|---|---|---|
| Driving lane | 4022 | 3633 | 899 | 700 | 435 | 9689 |
| Passing lane | 4877 | 3477 | 468 | 134 | 89 | 9045 |

### 2.6. Summary

In Section 2, the detailed methodology of FEM, machine learning, and the lateral distribution of wheel tracks using video analysis was explained. In FEM, the use of the detailed ABAQUS mechanism for analysis was explained. ML techniques were used to evaluate the most effective factor from multiple results derived from FEM. Bayesian optimization and SHAP interpretation mechanism were used. At the end of this section, the use of live video analysis to calculate the lateral distribution of load by wheel tracking was described.

## 3. Results

### 3.1. FEM Analysis

Using ABAQUS software, three scenarios were tested as mentioned in Table 3 above.

### 3.1.1. Scenario 1: With Speed 100 km/h

In this specific scenario, a speed limit of 100 km/h, action time (100,000 to 500,000), and temperature ranges (0 °C to 50 °C) were selected to assess the durability of the asphalt pavement. Using Equation (1), the duration of a single action time for axle loads was 0.00691 s, which was resolved while analyzing the results. It was determined that after 500,000 cycles of axle loading, the cumulative action time was 3456 s.

This is important in evaluating a pavement's ability to withstand repeated stress and strain. Figure 9 shows the temperature vs. rutting at a speed of 100 km/h in different action times.

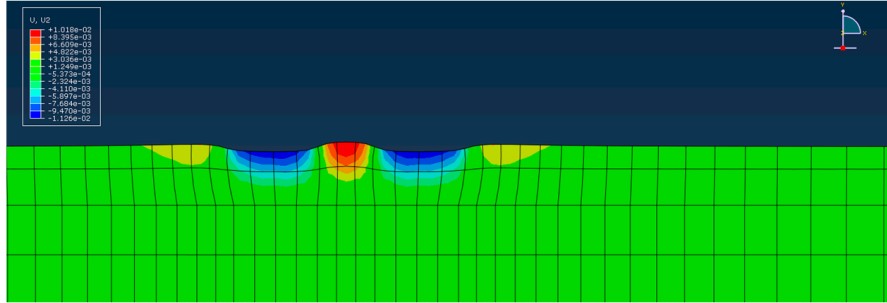

**Figure 9.** Rutting at 100 km/h.

From the graphs in Figure 10, the value of rutting has increased with the increase in action times and the increase from 7.83 mm to a value of 17.89 mm. It can be seen from Figure 10a when loading action time was 100,000 and temperature was 0 °C, maximum rutting was observed as 0.19608 mm, and rutting was observed to increase gradually when the temp was increased to 10 °C and later to 50 °C. Rutting value also increased from 0.19608 mm to 7.83 mm. Similarly in Figure 10b, when we increased the loading action time from 100,000 to 200,000, we observed a constant increment in rutting value. Rutting at 0 °C was observed 0.20825 mm and at 50 °C it was observed at 9.947 mm, around 15% more than previous simulation. Similarly, when we increased the loading action time to 500,000, a significant change in rutting was observed, that can be seen in Figure 10e. Maximum rutting at 50 degrees centigrade was observed 17.89 mm.

From this analysis it can be observed that when speed is constant at 100 km/h, and temperature is increased gradually from 0 to 50 °C, the graph of rutting increased.

### 3.1.2. Scenario 2: With Speed 80 km/h

In the second scenario, a speed limit of 80 km/h carried out 100,000 to 500,000 action times was imposed to determine the extent of rutting on the road surface. Figure 11 shows the temperature vs. rutting at a speed of 80 km/h in different action times.

**Figure 10.** Scenario 1: With speed at 100 km/h (**a**) temp vs. rutting at 100,000 action times, (**b**) temp vs. rutting at 200,000 action times, (**c**) temp vs. rutting at 300,000 action times, (**d**) temp vs. rutting at 400,000 action times, and (**e**) temp vs. rutting at 500,000 action times.

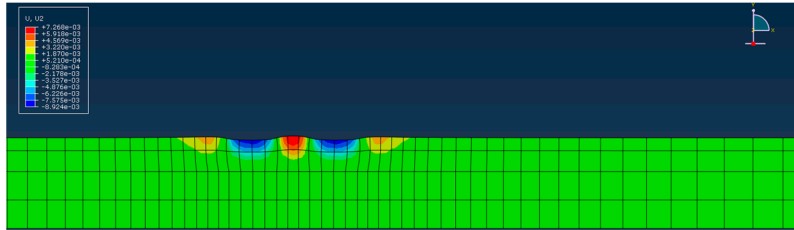

**Figure 11.** Rutting at 80 km/h.

In Figure 12, it can be seen that the value of rutting has increased with the increase of action times, an increase from 8.87 mm to a value of 21.24 mm. It can be seen from

Figure 12a when loading action time was 100,000 and temperature was 0 °C, maximum rutting of 0.1950 mm was observed; rutting was observed to increase gradually when we increased the temp to 10 °C and later to 50 °C. Rutting value also increased from 0.1950 mm to 8.87 mm. Similarly, in Figure 12b, when we increased the loading action time from 100,000 to 200,000, we observed a constant increment in rutting value. Rutting at 0 °C was observed to be 0.2185 mm and at 50 °C it was observed to be 10.7 mm, more than the previous simulation. Similarly, when we increased the loading action time to 500,000, a significant change in rutting was observed, that can be seen in Figure 12e. Maximum rutting at 50 degrees centigrade was observed 21.24 mm. From this analysis it can be observed that when speed is constant at 80 km/h and temperature increased gradually from 0 to 50 °C, the graph of rutting increased but rutting value is more than that of 100 km/h, which shows rutting is inversely proportional to the speed of the vehicle.

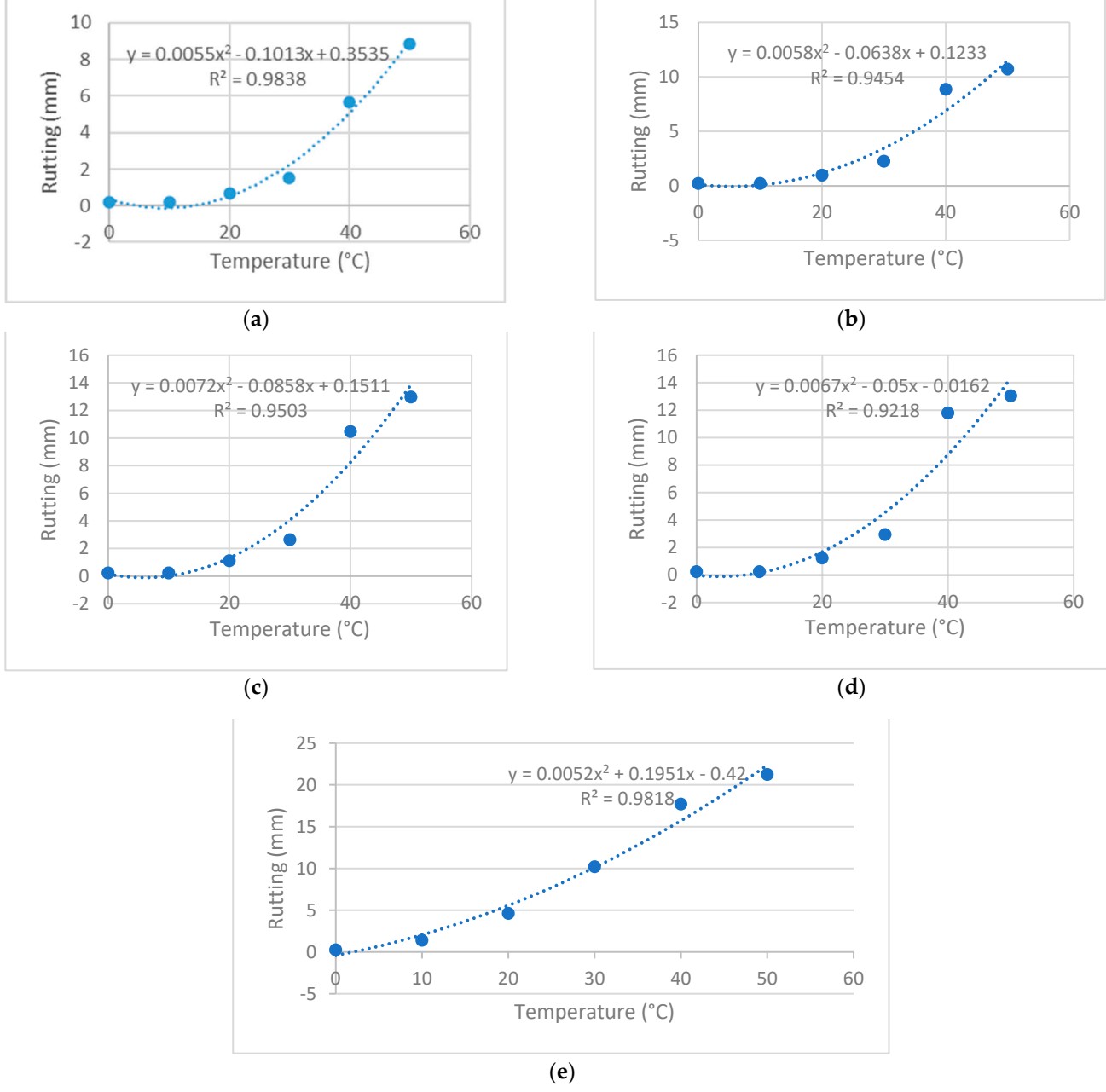

**Figure 12.** Scenario 2: With speed at 80 km/h (**a**) temp vs. rutting at 100,000 action times, (**b**) temp vs. rutting at 200,000 action times, (**c**) temp vs. rutting at 300,000 action times, (**d**) temp vs. rutting at 400,000 action times, and (**e**) temp vs. rutting at 500,000 action times.

### 3.1.3. Scenario 3: With Speed 60 km/h

In the third scenario, a speed limit of 60 km/h, and 100,000 to 500,000 action times were used to determine the action time and rutting of the road surface; Figure 13 shows the rutting at speed of 60 km/h.

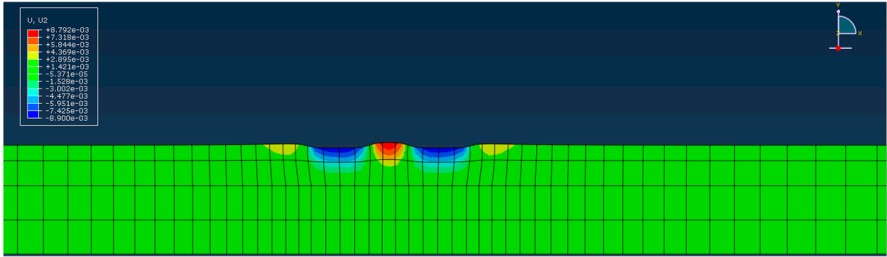

**Figure 13.** Rutting at 60 km/h.

In Figure 14, the value of rutting has increased with the increase of action times, an increase from 9.33 mm to a value of 24.7 mm. From this analysis of this scenario, it can be observed that when speed is constant at 60 km/h and temperature is increased gradually from 0 to 50 °C, the graph of rutting increased but the rutting value is more than that of 100 km/h and 80 km/h, which shows rutting is inversely proportional to speed of vehicle.

### 3.2. Prediction Analysis of Rut Depth Using ML Techniques

For this research, Python 3.7.1 was utilized to implement machine learning for the analysis of rut depth. The identification of optimal hyperparameters was carried out through Bayesian optimization, with details provided in Table 6.

**Table 6.** ML algorithm with hyperparameters, Range and optimal values.

| Models | Hyperparameters | Range | Optimal Values |
|---|---|---|---|
| LGBM | {(learning rate), (n_estimators), (num_leaves), (reg_lambda), (reg_alpha} | {(0.001–0.20), (200–2000), (30–100), (1.10–1.50), (1.10–1.50)} | {(0.15), (1179), (47), (1.18), (1.33)} |
| AdaBoost | {(learning rate), (n_estimators)} | {(0.001–0.20), (200–2000)} | (0.07), (350) |
| RF | {(n_estimators), (max_depth)} | {(200–2000), (2–16)} | (940), (11) |
| LR | {(learning rate)} | {(0.001–0.20)} | (0.05) |

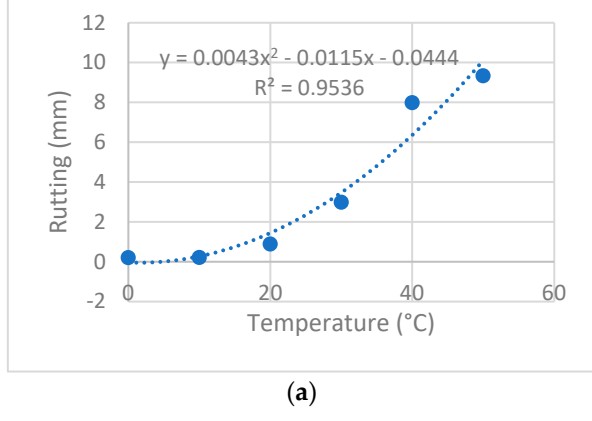

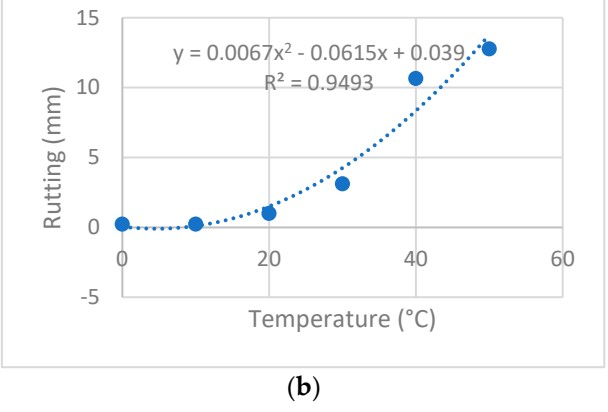

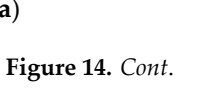

**Figure 14.** *Cont.*

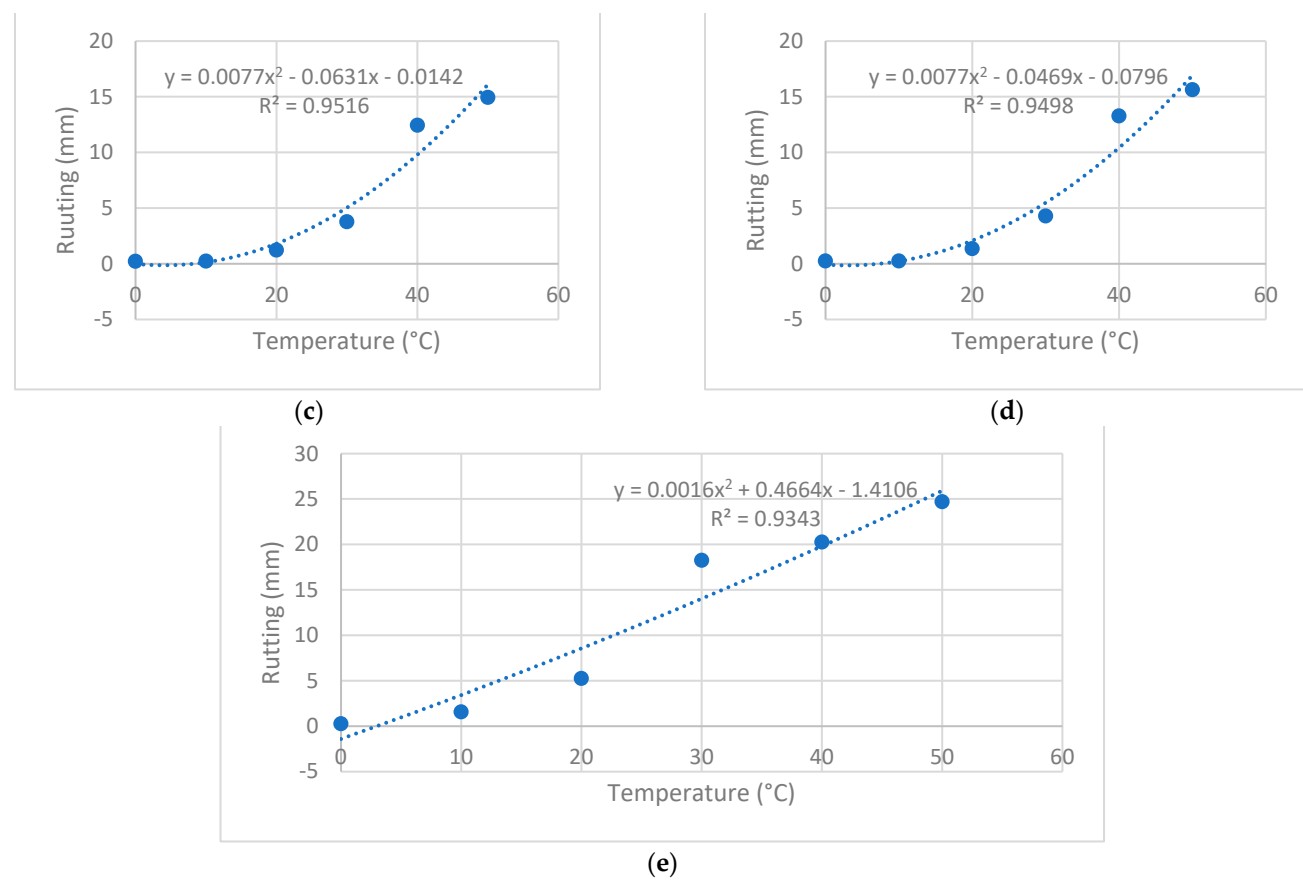

**Figure 14.** Scenario 3: With speed at 60 km/h (**a**) temp vs. rutting at 100,000 action times, (**b**) temp vs. rutting at 200,000 action times, (**c**) temp vs. rutting at 300,000 action times, (**d**) temp vs. rutting at 400,000 action times, and (**e**) temp vs. rutting at 500,000 action times.

To address overfitting, hyperparameter tuning was employed, and it is crucial to note the dataset underwent shuffling before partitioning (40% for training, 50% for testing). Post adjustment, metrics for model testing remained stable within a 95% confidence interval. The LGBM model outperformed with MAE (0.460), MSE (0.903), RMSE (0.950), and $R^2$ (0.977), while the RF model exhibited less favorable metrics (MAE 1.091, MSE 4.247, RMSE 2.061, $R^2$ 0.894), confirmed by graphical assessments in Figure 15. Table 7 shows the performance measures.

**Table 7.** Performance of training and testing dataset using ML.

| Models | Training Dataset | | | | Testing Dataset | | | |
|---|---|---|---|---|---|---|---|---|
| | **MAE** | **MSE** | **RMSE** | **$R^2$** | **MAE** | **MSE** | **RMSE** | **$R^2$** |
| AdaBoost | 1.695 | 4.512 | 2.124 | 0.881 | 1.489 | 3.566 | 1.888 | 0.911 |
| LGBM | 0.335 | 0.481 | 0.694 | 0.987 | 0.460 | 0.903 | 0.950 | 0.977 |
| RF | 1.274 | 5.326 | 2.307 | 0.859 | 1.091 | 4.247 | 2.061 | 0.894 |
| LR | 1.315 | 4.420 | 2.102 | 0.882 | 1.275 | 4.211 | 2.05 | 0.894 |

### 3.2.1. LGBM Model Interpretation by SHAP

After evaluating the $R^2$ value on the testing dataset, LGBM was chosen to model rutting concerning the specified factors. This section demonstrates the comprehensive and detailed insights obtained through SHAP analysis, including the primary and interactive effects of the factors.

### 3.2.2. Global Factor Interpretation

The global interpretation involves assessing SHAP factor importance and contribution, as illustrated in Figure 16. In (a), the mean absolute SHAP values highlight "Temperature (T)" as the most influential factor (2.874), followed by "No. of load (N)" (1.157), and "No. of wheels (nw)" (1.102). In (b), the bee swarm plot color-coded for factor values underscores the impact, with "Temperature (T)" in red, suggesting a higher likelihood of increased rutting, while "No. of wheels (nw)" in blue indicates a significant effect on rutting.

**Figure 15.** Analysis of (**a**) AdaBoost model, (**b**) LGBM model, (**c**) RF model, (**d**) LR model.

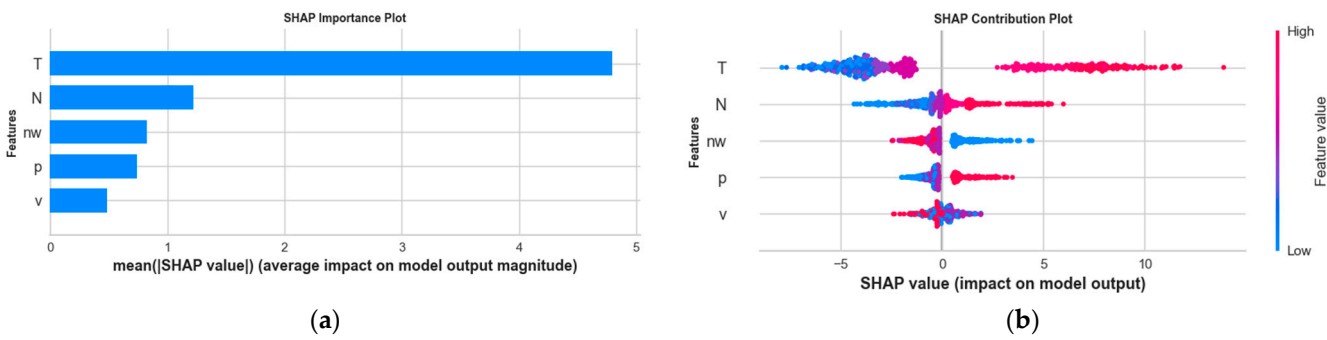

**Figure 16.** Global Factor Interpretation; (**a**) factor importance plot; (**b**) factor bee swarm plot.

### 3.2.3. Single Factor Analysis

The depiction plot of SHAP (Figure 17) serves to understand the influence of a singular significant factor on the LGBM prediction model's output. Alterations in plot values allow the observation of changes in the factors' relative importance. SHAP values surpassing zero indicate the potential impact on rutting intensity for specific factors. Figure 17a–c elucidates the influence of three notable factors—namely, "Temperature ($T$)", "No. of Load Cycles ($N$)", and "No. of Wheels ($nw$)" on rut depth. Elevated temperatures exceeding 30 °C and increased loading cycles contribute to heightened rutting, while a lower count of wheels results in intensified rutting.

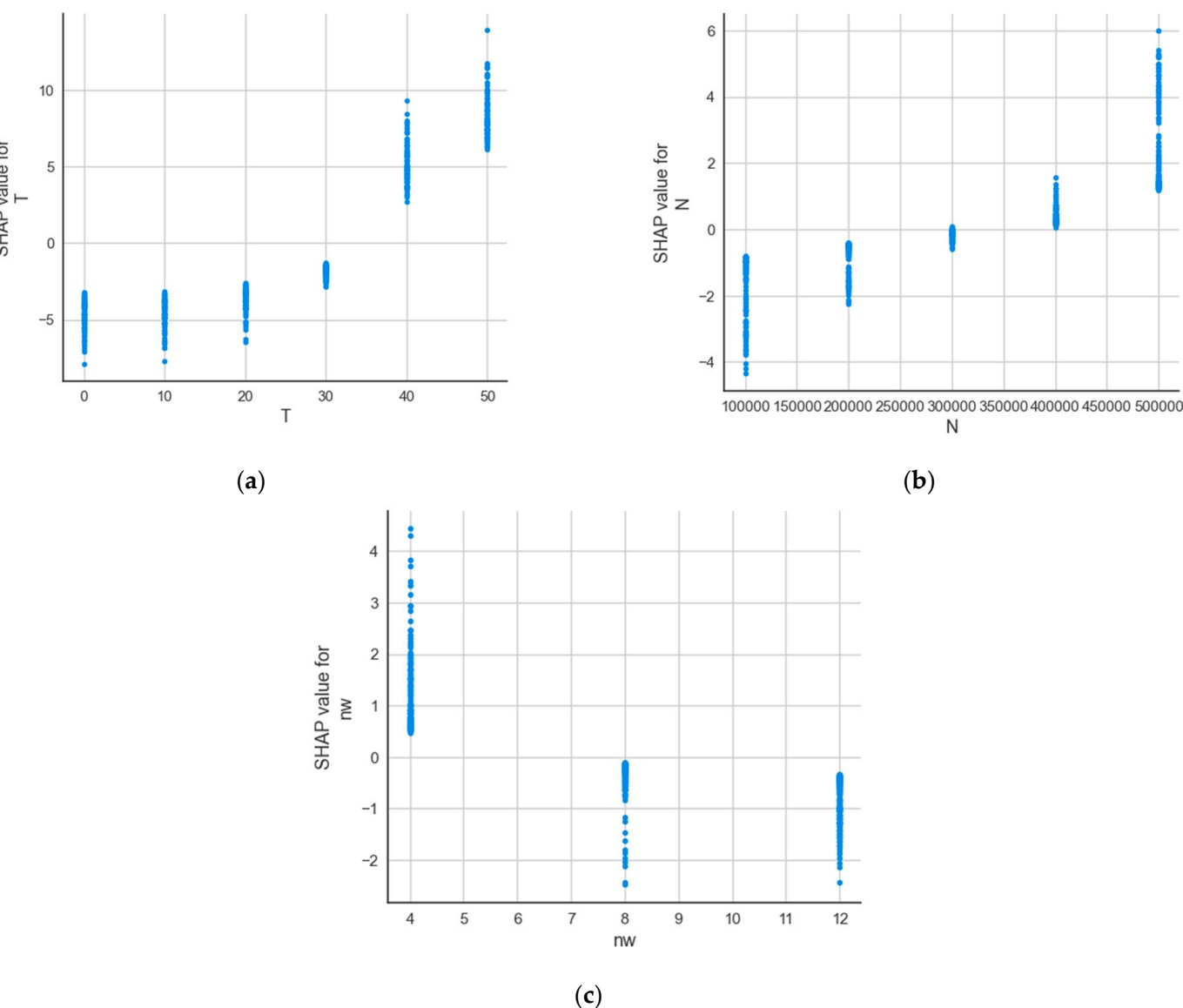

(**a**)

(**b**)

(**c**)

**Figure 17.** Single factor analysis: (**a**) effect of temperature, (**b**) effect of number of load, (**c**) effect of number of wheels.

### 3.2.4. Factor Interaction Analysis

Evaluation of SHAP interaction plots (depicted in Figure 18) aims to understand the interplay between the factors utilized for assessing the optimal RF forest, specifically examining their collective contributions and interactions. Figure 18a demonstrates higher rutting intensity in the presence of temperature and number of loading cycles, with temperature values above 30 °C and a loading cycle above 300,000 causing higher rutting.

Figure 18b demonstrates the comparison between the number of wheels and temperature, with higher temperature and lower number of wheels causing high damage in the form of rutting. In addition, Figure 18c is the representation between the number of wheels and the number of loading cycles, with higher value of $N$ and lower value of $n_w$ causing more rutting in the pavements.

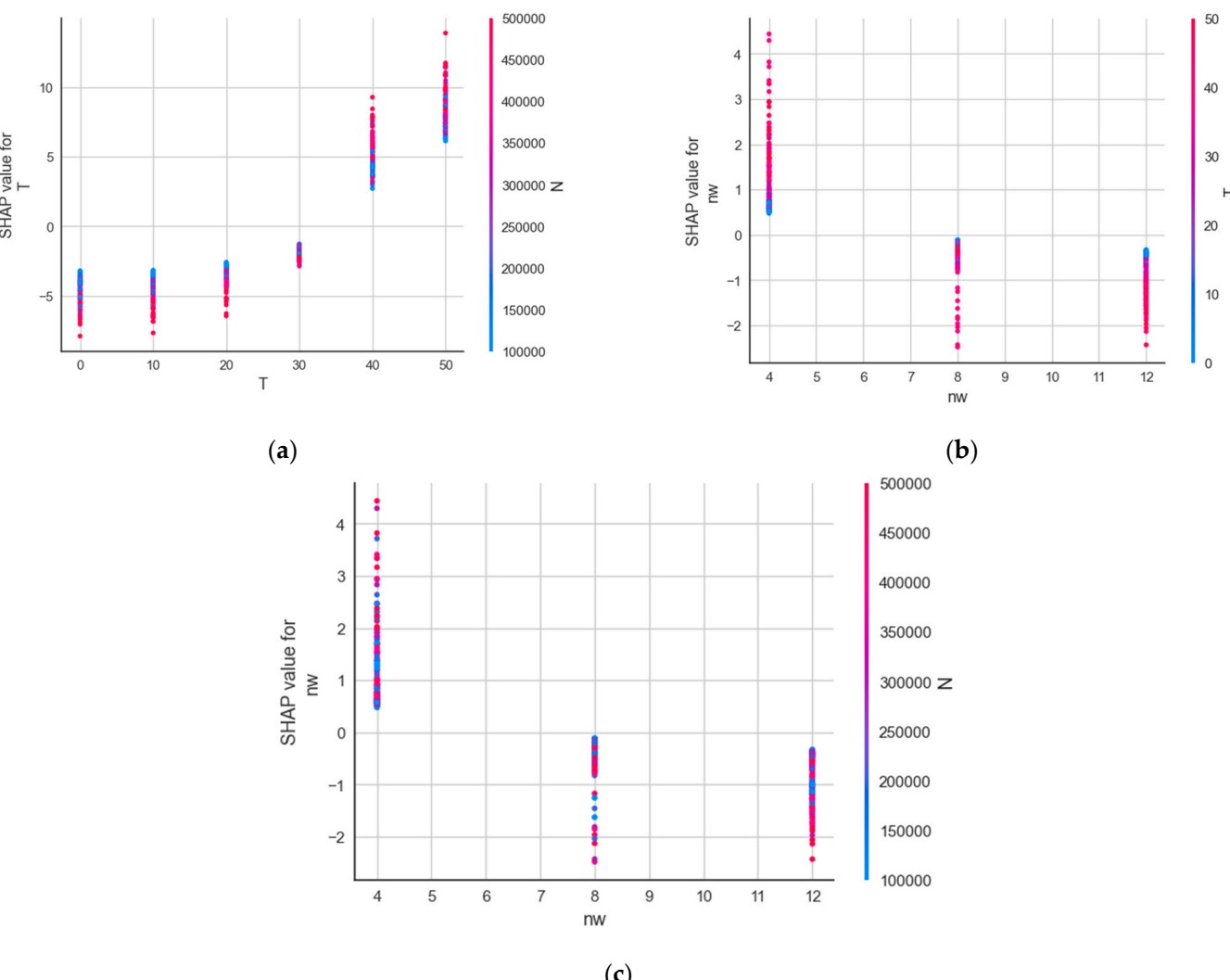

(**a**)

(**b**)

(**c**)

**Figure 18.** Interaction analysis: (**a**) effect of temperature and number of loads, (**b**) effect of number of wheels and temperature, (**c**) effect of number of wheels and number of loads.

### 3.3. Lateral Distribution of Wheel Tracks

#### 3.3.1. Lateral Distribution of Wheel Tracks of Small Cars

Small vehicles exhibit enhanced maneuverability and a scattered wheel track distribution due to their compact width, providing greater freedom within the lane. However, the prevalence of larger and heavy-duty vehicles affects small vehicles' freedom. In the overtaking lane, small vehicles, being more numerous, show a distribution linked to their faster speeds. The bimodal Gaussian curve in Figure 19a illustrates the distribution of wheels track of compact vehicles, displaying a broader range and diminished peak value. In the overtaking lane (Figure 19b), the swift small vehicles exhibit a slightly wider distribution range, with coverage rates in both the left (77%) and right (83%) distribution of wheels track peak areas.

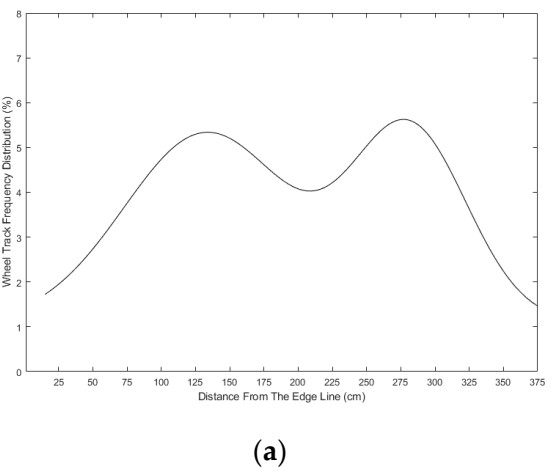

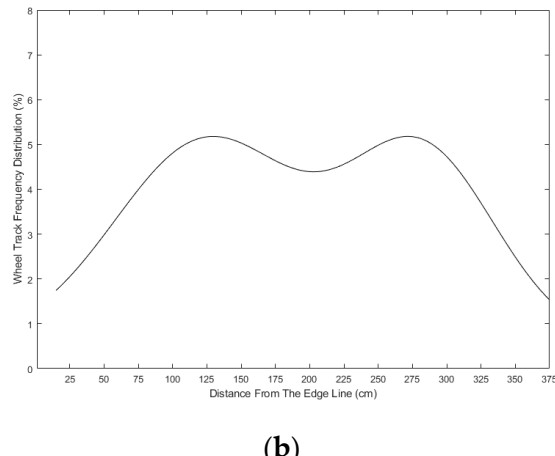

(**a**)　　　　　　　　　　　　　　　　　(**b**)

**Figure 19.** The distribution of wheels track of small cars in (**a**) driving lane, (**b**) passing lane.

### 3.3.2. Lateral Distribution of Wheel Tracks of Medium Trucks

Medium-sized trucks, being wider and slower than small cars, exhibit more concentrated wheel tracks, limiting their freedom within the lane. Figure 20a illustrates a fitting curve, revealing a larger peak value and slightly narrower distribution range compared to small cars. The left and right wheel peak areas have coverage rates of 70% and 71%, respectively. In the overtaking lane, where medium-sized trucks coexist with faster-moving vehicles, their wheel track distribution is slightly scattered (Figure 20b). With a slightly diminished peak value, the dispersion range expands compared to the driving lane. Left and right wheel track distribution peaks boast coverage rates of 75% and 74%, respectively.

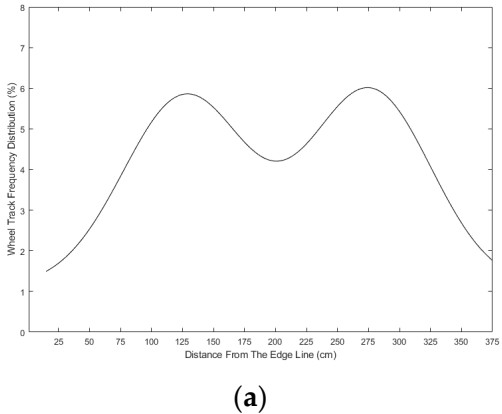

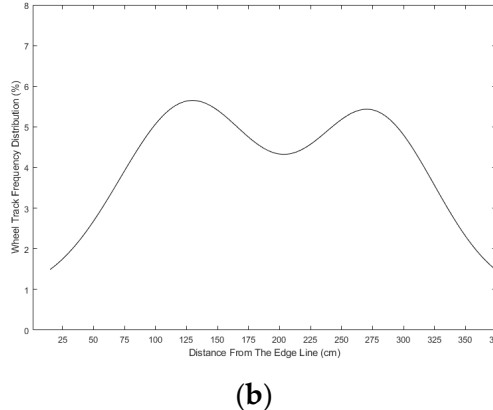

(**a**)　　　　　　　　　　　　　　　　　(**b**)

**Figure 20.** Wheel track distribution of medium sized truck in (**a**) driving lane, (**b**) passing lane.

### 3.3.3. Lateral Distribution of Wheel Tracks of Large Buses

Large buses, characterized by extended bodies, wider widths, and slower speeds, tend to maintain a centralized position within the lane for more restricted driving freedom. In comparison to smaller vehicle models, the wheel tracks of large buses exhibit greater concentration, with a larger peak value and a narrower distribution range (Figure 21a). In both driving and overtaking lanes, large buses consistently favor the center, resulting in similar wheel track distributions. The Gaussian bimodal fitting curve (Figure 21b) indicates comparable peak values and distribution ranges between the two lanes, with approximately 68% coverage in left and right areas.

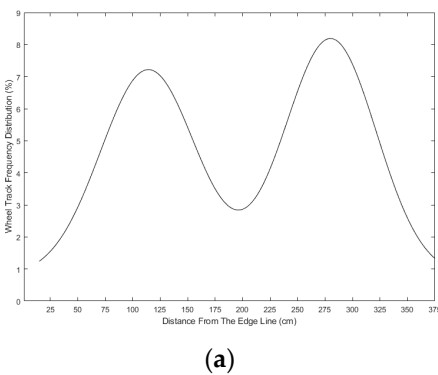

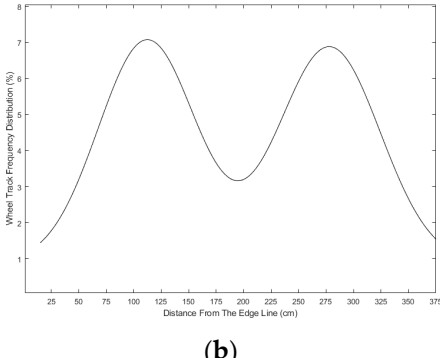

(**a**)

(**b**)

**Figure 21.** Wheel track distribution of large buses in (**a**) driving lane, (**b**) passing lane.

### 3.3.4. Lateral Distribution of Wheel Tracks of Large Trucks

Large trucks, characterized by their substantial load capacity and slower speed, exhibit constrained driving freedom and a more concentrated wheel track distribution. In Figure 22a, the fitting curve diagram illustrates larger peak values and narrower distribution ranges in large trucks. Both side peak areas show coverage rates of 65% and 66%, respectively. Despite the relatively small number of big trucks in the fast lane, their inherent limitations lead to a significant concentration of wheel tracks, as depicted in the Gaussian bimodal fitting curve in Figure 22b. The peak values and distribution ranges in the wheel track peak area remain comparable to those observed in the driving lane, with consistent coverage rates of 65% and 66% in both sides of the lanes.

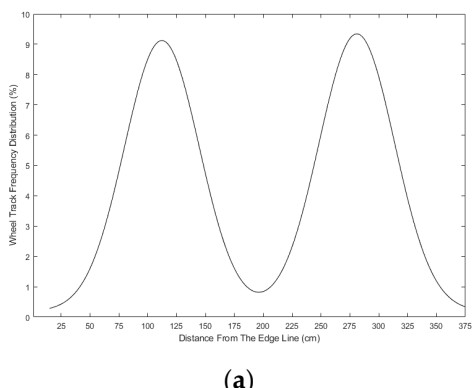

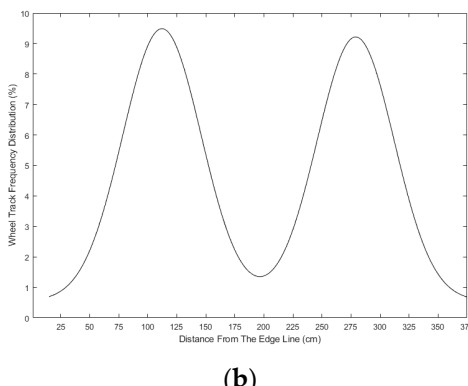

(**a**)

(**b**)

**Figure 22.** Wheel track distribution of large trucks in (**a**) driving lane, (**b**) passing lane.

### 3.3.5. Lateral Distribution of Wheel Tracks of Trailers

Trailers, characterized by their wide and long bodies, limited maneuverability, and slow speed, exhibit a fixed and centered driving trajectory within lanes. This leads to a pronounced concentration of wheel track distribution, with the trailer displaying the most substantial peak value and a narrow distribution range, as illustrated in Figure 23a. In both the driving and overtaking lanes, the wheel track distribution of trailers remains similar due to their limited presence in the overtaking lane, as depicted in the Gaussian bimodal fitting curve shown in Figure 23b. Notably, the distribution in both lanes share coverage rates of 61% and 62%, respectively.

### *3.4. Summary*

This chapter was divided into three sections for the results; firstly, the three scenarios tested using FEM were described. It is concluded from the results that rutting is directly proportional to temperature and loading cycles but opposite in the case of vehicle speed. Secondly, four machine learning techniques were used to predict the most effective factor among others and temperature had the highest effect in causing damage to road structure.

Subsequently, the lateral distribution of wheel tracks was explained by video analysis, and different vehicle loading frequencies on different sections of the road were explained.

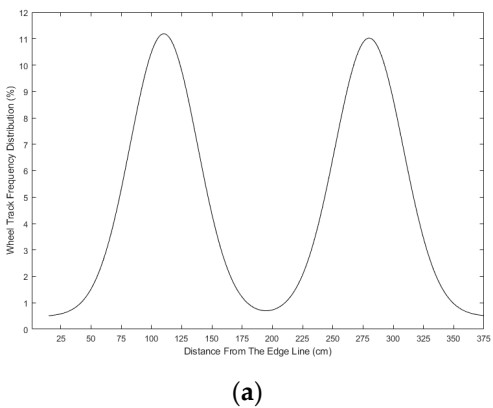

(**a**)

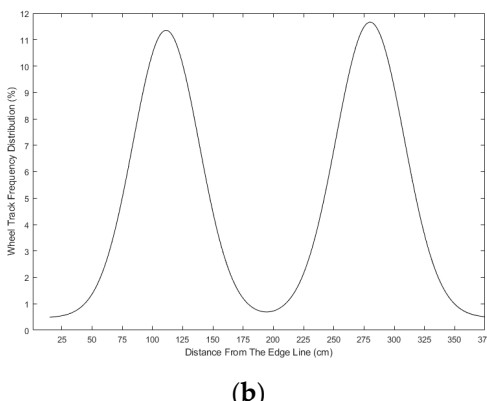

(**b**)

**Figure 23.** Wheel track distribution of trailers in (**a**) driving lane, (**b**) passing lane.

## 4. Conclusions

This study investigated the impact of temperature, vehicle speed, and load cycles on asphalt pavements, revealing those carrying vehicles with lower speeds correlate with higher rutting values due to prolonged contact time. Deformation is notably affected by load cycles, suggesting that increased temperatures and reduced speeds amplify pavement deformation. The findings stress the importance of designing pavements considering varying temperatures throughout the year. The conclusion of the video analysis showed that the diverse wheel track distributions observed among small vehicles, medium-sized trucks, large buses, large trucks, and trailers reflect distinct characteristics influencing their maneuverability and driving freedom within lanes. Small vehicles, with their compact width providing enhanced maneuverability, demonstrated scattered wheel tracks, particularly in the overtaking lane where their faster speeds prevail. Medium-sized trucks exhibited more concentrated wheel tracks, and limited freedom within the lane due to their wider width and slower speed. Large buses maintained a centralized position within both lanes, showing a concentrated wheel track distribution, while large trucks, with substantial load capacity, exhibited constrained driving freedom and concentrated wheel tracks. Trailers, with wide and long bodies, displayed a fixed and centered driving trajectory, leading to a pronounced concentration of wheel tracks. This analysis provides valuable insights into the distinct lane behavior of various vehicle types, offering considerations for traffic management, lane design, and overall road infrastructure planning.

Additionally, interpretable machine learning models, particularly the Bayesian-optimized LGBM model, outperformed the RF model in rut depth prediction. SHAP interpretation identified temperature and loading frequency as influential factors, enhancing our understanding of rutting formation, and empowering engineers with insights for targeted mitigation strategies. The results underscore temperature's pivotal role in pavement performance, emphasizing its consideration in design and management practices.

**Author Contributions:** X.H., Conceptualization, methodology, software, validation, formal analysis, funding acquisition; A.I., resources, data curation, writing—original draft preparation, writing—review and editing; A.K., visualization, supervision, software; F.C., project administration, supervision. All authors have read and agreed to the published version of the manuscript.

**Funding:** This research received no external funding.

**Institutional Review Board Statement:** Not applicable.

**Informed Consent Statement:** Not applicable.

**Data Availability Statement:** Data are contained within the article.

**Acknowledgments:** We are grateful to all friends and professors who helped to collect the video from Pakistani roads, weather data from meteorological departments, and helped to analyze the video data at the College of Transportation Engineering Tongji University.

**Conflicts of Interest:** The authors declare no conflicts of interest.

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
