# Peer review of "Assessment of Factors Affecting Pavement Rutting in Pakistan Using Finite Element Method and Machine Learning Models"

_sustainability, doi:10.3390/su16062362_

Round 1
Reviewer 1 Report
Comments and Suggestions for Authors
1. Title is suggested to be revised: "Assessment of Factors Affecting Pavements Rutting in Pakistan using Finite Element Method and Machine Learning Models" . Reason: The revised title suggests that Machine Learning and Finite Element Method are parallel approaches rather than Machine Learning merely assisting.
2. Keyword Suggestions: Add 'rutting', remove 'ABAQUS' and 'MATLAB'.
3. Table 1 provides parameters such as Thermal conductivity, Heat capacity, etc., but these parameters do not seem to be mentioned in the research. What is the use of these parameters?
4. Table 4: It needs to be explained whether different vehicle types were considered when mentioning No of wheels (nw), whether they are all truck models, or also include small passenger cars, which have completely different effects on road ruts.
5. Section 2.5.1: Provide details on the number of cameras used, their placement intervals, and the total amount of data collected.
6. Figure 9: What is U2. How is the value of ruts determined through FEM graph like this?
7. 3.3: This section on the lateral distribution of wheel tracks is too complicated. It is recommended not to divide it into sub-sections. Instead, it suggests to summarize Figures 19 to 23 together to analyze the lateral distribution of different shapes in a unified manner. The specific data can be expressed in tables. In addition, it is recommended to analyze the impact of these distributions on rutting, such as using lateral distribution + different vehicle action frequencies + FEW + ML to evaluate and predict rutting. This will make more sense for research.
8. It is recommended to summarize the important innovative conclusions obtained from the each parts, including finite element analysis, machine learning, and lateral distribution.
Comments on the Quality of English LanguageThe English expression in writing needs to be carefully revised to meet the details. There are many problems that exist, and the following are some (but not all):
(1) Acronyms should be fully spelled out when first used, such as WBM, CCTV etc.
(2) Table 1:
Missing units for parameters.
"Water stabilized base" should be "Cement stabilized base"?
2.041092x10-4 should be written as 10 to the power of -4.
(3) All mathematical symbols in the text should be italicized, such as T, t, nw, N, etc.
(4) Table 2: In the Value column, units should not be repeated. KN should be kN.
(5) Lines 198-205: An explanation of the corresponding symbol is required. Should 1.2.3 be repeated with the first level title sequence number.
(6) Lines 243-248, 252-254: 'Where' should not be followed by a semicolon. Lack of full stops of each explanation. Incomplete parentheses of (PDF).
(7) Figure 8: "identificatio" should be corrected to "identification".
(8) Author Contributions: author names should be provided.
Reviewer 2 Report
Comments and Suggestions for Authors
1. The abstract should be refined to explicitly claim the advantages and innovations of the proposed method and the concrete performance of the method.
2. The figures in the experimental section are chaotic and confusing. Some of them are large and some of them are small. In addition, the text in some figures is particularly small and difficult to read.
3. The titles of some chapters are inappropriate.
4. Please consider to add the following references:
Zhiquan Liu, Lin Wan, Jingjing Guo, Feiran Huang, Xia Feng, Libo Wang, and Jianfeng Ma, PPRU: A Privacy-Preserving Reputation Updating Scheme for Cloud-Assisted Vehicular Networks, IEEE Transactions on Vehicular Technology, 2023.
Jingjing Guo, Xinghua Li, Zhiquan Liu, Jianfeng Ma, Chao Yang, Junwei Zhang, Dapeng Wu, TROVE: A Context Awareness Trust Model for VANETs Using Reinforcement Learning, IEEE INTERNET OF THINGS JOURNAL, 2020
Comments on the Quality of English LanguageThe language must be further improved to make the whole manuscript easier to follow. In addition, there are some grammar mistakes and typos must be fixed.
Reviewer 3 Report
Comments and Suggestions for Authors
In this manuscript, a method combining FEM with machine learning is proposed to analyze the impact of environmental factors, vehicle dynamics, and loading conditions on pavement structures. This manuscript is well-organized, and it can be accepted after minor revision. Specific comments are as follows:
(1) Please provide the full names for the first occurrence of any English acronyms in the article. After this initial mention, use the acronym directly. Review and correct instances where this has been overlooked.
(2) In Section 1, there should be an overview of the content of this study. Add an introduction to related research in this field and highlight the innovation and improvements of this study relative to others. Also, consider adding a brief introduction to the structure of the entire paper at the end of this section.
(3) In Section 2.2, a more detailed description should be added for Figure 2.
(4) Images and tables should be displayed on the same page, as seen with Table 1, Table 7, etc. Check for and amend such issues.
(5) In Figure 3, within the Inputs, the expressions "No of load" and "No of wheels" are incorrect. It should be "No." instead of "No". Carefully review and correct spelling and formatting errors throughout the paper.
(6) More recent works about machine learning and load analysis should be included in this manuscript, like “Fatigue reliability analysis of composite material considering the growth of effective stress and critical stiffness. Aerospace, 2023, 10(9): 785.”, “A novel machine learning method for multiaxial fatigue life prediction: Improved adaptive neuro-fuzzy inference system. International Journal of Fatigue, 2024, 178: 108007.”
(7) In Section 3.2, Table 6's hyperparameter choices have many Optimal Values outside the parameter Range. Is this reasonable? Provide an explanation and correct this issue.
(8) The font size in the images is too small to read comfortably. Increase the font size in images, as in Figure 15.
Reviewer 4 Report
Comments and Suggestions for Authors
The topic is interesting. However, I don't find anything new. The conclusions are evident. I do not see any contribution to the state of knowledge on the topic of study. The bibliographic review consulted is very scarce. I do not find in the manuscript a deep analysis or discussion of the results. Based on the above, I consider that the article should be rejected. Some additional concerns are: What does WBM mean? Why authors only simulate one structure? What would be the effect if the structure changed the layer thicknesses? What happens when the vehicle speed is less than 60 km or when the tire ground pressure varies with respect to 0.7 MPa? 100,000 to 500,000 charge cycles is very small. Why did you choose that range and what would the pavement performance be like with higher cycles? What is the constitutive equation that you used to model the asphalt and granular layers? It is not clear how they estimate the ground pressure of tire a priori. It is not clear how the "No of wheels" parameter is simulated in the FEM, nor how they do the meshing. What is the difference of the study with respect to other similar ones?
Round 2
Reviewer 4 Report
Comments and Suggestions for Authors
The authors made changes to the manuscript. They took my comments into account.